# The Impact of General Anesthesia on Redox Stability and Epigenetic Inflammation Pathways: Crosstalk on Perioperative Antioxidant Therapy

**DOI:** 10.3390/cells11121880

**Published:** 2022-06-09

**Authors:** Stelian Adrian Ritiu, Alexandru Florin Rogobete, Dorel Sandesc, Ovidiu Horea Bedreag, Marius Papurica, Sonia Elena Popovici, Daiana Toma, Robert Iulian Ivascu, Raluca Velovan, Dragos Nicolae Garofil, Dan Corneci, Lavinia Melania Bratu, Elena Mihaela Pahontu, Adriana Pistol

**Affiliations:** 1Clinic of Anaesthesia and Intensive Care, Emergency County Hospital “Pius Brînzeu”, 300723 Timișoara, Romania; stelian.ritiu@umft.ro (S.A.R.); dsandesc@yahoo.com (D.S.); bedreag.ovidiu@umft.ro (O.H.B.); marius.papurica@gmail.com (M.P.); popovici.sonia@yahoo.com (S.E.P.); daiana.toma@yahoo.com (D.T.); raluca_velovann@yahoo.com (R.V.); 2Faculty of Medicine, “Victor Babeș” University of Medicine and Pharmacy, 300041 Timișoara, Romania; bratu.lavinia@umft.ro; 3Anaesthesia and Intensive Care Research Center (CCATITM), “Victor Babeș” University of Medicine and Pharmacy, 300041 Timișoara, Romania; 4Faculty of Medicine, “Carol Davila” University of Medicine and Pharmacy, 050474 Bucharest, Romania; ivascu.robert.iulian@gmail.com (R.I.I.); dcorneci@yahoo.com (D.C.); adriana.pistol@insp.gov.ro (A.P.); 5Clinic of Anaesthesia and Intensive Care, Central Military Emergency Hospital “Dr. Carol Davila”, 010242 Bucharest, Romania; 6Faculty of Pharmacy, “Carol Davila” University of Medicine and Pharmacy, 050474 Bucharest, Romania; elena.pahontu@umfcd.ro

**Keywords:** general anesthesia, redox, inflammation, antioxidants, hypermetabolism, microRNAs, biomarkers, oxidative stress, vitamin C

## Abstract

Worldwide, the prevalence of surgery under general anesthesia has significantly increased, both because of modern anesthetic and pain-control techniques and because of better diagnosis and the increased complexity of surgical techniques. Apart from developing new concepts in the surgical field, researchers and clinicians are now working on minimizing the impact of surgical trauma and offering minimal invasive procedures due to the recent discoveries in the field of cellular and molecular mechanisms that have revealed a systemic inflammatory and pro-oxidative impact not only in the perioperative period but also in the long term, contributing to more difficult recovery, increased morbidity and mortality, and a negative financial impact. Detailed molecular and cellular analysis has shown an overproduction of inflammatory and pro-oxidative species, responsible for augmenting the systemic inflammatory status and making postoperative recovery more difficult. Moreover, there are a series of changes in certain epigenetic structures, the most important being the microRNAs. This review describes the most important molecular and cellular mechanisms that impact the surgical patient undergoing general anesthesia, and it presents a series of antioxidant therapies that can reduce systemic inflammation.

## 1. Introduction

Anesthesia allows the performing of surgical procedures in a rapid, safe, and pleasant manner, producing analgesia, absence of awareness, and adequate muscle relaxation when needed [1,2,3]. A critical aspect of perioperative anesthetic care is the maintenance of homeostasis, including hemodynamic stability, oxygenation, ventilation, and temperature. The World Health Organization (WHO, www.who.int/, accessed on 20 May 2022) and the World Bank (WB, www.worldbank.org, accessed on 20 May 2022) expect that by 2026, the burden of diseases requiring surgery and anesthesia will exceed that of HIV, tuberculosis, and malaria, measured in disability-adjusted life years. As anesthesia providers are an integral part of the delivery of safe and effective surgical care, it is imperative to develop the necessary tools to minimize mortality and morbidity in the perioperative period [4,5].

Surgical stress induces an immuno-inflammatory response not necessarily proportional to the degree of tissue damage. In addition to the surgical injury, other factors may be contributing to the inflammatory response in the perioperative period, including the general anesthesia itself, mechanical ventilation, the administration of blood products, and antiemetic drugs [6]. Moreover, numerous studies have highlighted a number of molecular changes induced by anesthetic substances, involving the expression of inflammation and hypermetabolism during general anesthesia. An abnormal over- or under-expressed immuno-inflammatory response has been associated with various relevant postoperative conditions, including infections, pulmonary complications, delirium and postoperative cognitive dysfunction [7], renal injury [8], and cancer recurrence [9,10,11]. Another important molecular and cellular effect is the unbalance of the expression of antioxidants in relation to the activity of reactive oxygen and nitrogen species (ROS, RNS). This phenomenon is called oxidative stress (OS) and refers to redox activity or reduction–oxidation activity [9,10].

This review article aims to integrate current knowledge regarding the effects of commonly used anesthetic agents on the inflammatory response developed in the postoperative period.

## 2. Redox Disturbance and Inflammation during General Anesthesia and Surgery Procedures

A degree of inflammation during surgery is unavoidable and represents the first necessary mechanism in wound healing. It involves complex pathways that alter endocrine, hemodynamic, metabolic, redox, and immune responses [12]. The main link between localized injury and a systemic inflammatory response is represented by the damage-associated molecular pattern (DAMP) molecules, also called alarmins [13]. These molecules have a physiological role inside the cell but acquire additional functions when exposed to the extracellular environment: they can activate antigen-presenting cells (APCs); have chemotactic properties; and exhibit immunoenhancing activity, stimulating both the innate and adaptive immune system [14]. Alarmins include high mobility group box 1 (HMGB1), heat shock proteins (HSPs), defensins, cathelicidin, eosinophil-derived neurotoxin (EDN), S100 proteins, purine metabolites, and DNA or RNA located outside the nucleus or mitochondria. Hence, alarmins represent a diverse and structurally different group of molecules released by damaged or dying cells [15]. Interestingly, these “danger”-sensing molecules exert their functions by interacting with specific receptors expressed on damaged or dying cells but not on apoptotic cells [16].

As the first line of defense following tissue injury, migrating macrophages and granulocytes produce a pro-inflammatory reaction at the site of injury, which stimulates the release of various pro-inflammatory cytokines [17,18,19]. This response is rapidly contained by an anti-inflammatory response that aims to fine-tune and produce a balance between the defense and healing processes [20].

From a biochemical viewpoint, oxidative stress is characterized by an imbalance between oxidant and antioxidant factors, although pro-oxidative species have a higher share. The most reactive of all these species are hydrogen peroxide, hydroxyl radicals, superoxide anions, reactive nitrogen species, reactive lipid species, and oxidative fragments resulting from protein denaturation [21,22,23]. Free radicals can be formed both intra- and extracellularly due to certain exogen factors that manage to penetrate human biological systems. The most common intracellular sources of oxidant substances are nicotinamide adenine dinucleotide phosphate (NADPH) oxidase, mitochondria, the endoplasmic reticulum, lysosomes, cytochrome P450, and peroxisomes [24]. The body has a series of enzymatic systems, such as superoxide dismutase (SOD), glutathione peroxidase (GPX), and catalase (CAT), as well as certain antioxidant molecules, such as glutathione (GSH), vitamin E, vitamin C, melatonin, uric acid, and a series of polyphenols, with important antioxidant properties. The most well-known and widely researched oxidative reduction mechanism is that of SOD, responsible for the molecular conversion of superoxide anions to hydrogen peroxide, which leads to the formation of water through enzymatic catalyzation induced by CAT and GPX [25,26]. The reduction of oxidative species by GSH with the formation of glutathione disulfide (GSSG) is another mechanism of oxidative reduction [27]. Furthermore, reactive oxygen species help modify the cellular signaling pathways responsible for cellular proliferation, differentiation, autophagia, and apoptosis. Responsible for this is the activation of pro-inflammatory mechanisms, which leads to the formation of cytokines by activating nuclear factor κB (NF-κB), AMP-activated protein kinase, hypoxia-inducible factor, and Kelch-like ECH-associated protein 1 (KEAP1) [28]. During pro-oxidative states, at a cellular level, the mitochondria are responsible for generating oxidative factors involved in the augmentation of the redox state, which has further molecular and cellular effects. The mechanism is known as mitochondrial reactive oxidative species production (mitoROS) [29,30]. In the mitochondria, complexes I, II, and III are involved in the production of reactive oxygen species. From a molecular point of view, complex I is responsible for the entrance of electrons from the NADH complex into the respiratory chain. Therefore, the interaction of flavinmononucleotides with oxygen will result in the hydroxyl anion that will afterward be released in the mitochondrial matrix [31,32,33]. Complex II helps produce reactive species through the oxidation of succinate to fumarate, a part of the Krebs cycle. Increased amounts of ROS are produced by complex II when complexes I and III are blocked at a molecular level. Following redox activity, complex III leads to the production of high amounts of hydrogen peroxide, which is able to diffuse in the mitochondrial matrix [34,35]. Once the biological systems responsible for the adequate functioning of the mitochondria have been pro-oxidatively damaged, increased amounts of ROS will lead to mitochondrial death and the accumulation of highly oxidative species inside the cell. Physiologically, the mechanisms responsible for inhibiting the pro-oxidative destructive mechanisms inside the cell are GSH, thioredoxin (Trx), and glutaredoxin (Grx) [36]. Recent studies have shown increased oxidative reduction activity inside the mitochondria for SOD, which functions by blocking the redox activity of oxygen ions in hydrogen peroxide. According to Ribas et al., during enzymatic antioxidant activity in the mitochondria, the most important activity is that of SOD2 (manganese-dependent superoxide dismutase, MnSOD) [37], while in the intermembrane space, it is that of SOD1 (Cu, Zn-SOD). Through the activity of the SOD molecular family, reactive oxygen species are transformed into hydrogen peroxide, which has a less pronounced oxidative character. Enzymatic antioxidant species such as glutathione reductases, peroxidases, and peroxiredoxins are activated for the total inhibition of redox activity. They will dissociate hydrogen peroxide into water and oxygen. Regarding the Trx and Grx antioxidant enzymes, researchers have discovered different species, such as Trx2 and thioredoxin reductase-2 (TrxR2), responsible reducing oxidative stress by modulating NADPH mechanisms and taking control of electron migration inside the mitochondria [38,39,40,41]. The family of Grx antioxidant enzymes includes Grx2 and Grx5, responsible for modulating molecular mechanisms that produce oxygen ions in complex I by catalyzing thiol groups in GSH [42]. An increase in the redox state at the cellular organelle level will lead to a decrease in ATP production due to the migration of electrons toward the water used in the formation of hydrogen peroxide. Other important molecular sources of oxidative species inside the cell are characterized by the interaction between reactive oxygen ions with D-aspartate oxidase, xanthine oxidase, polyamine oxidase [43,44], (ACOX), L-αhydroxyacid oxidase, D-amino acid oxidase, and L-pipecolic oxidase. Due to insufficient electrons for ATP production at the mitochondrial level, the whole cell will be affected by the so-called energy failure phenomenon, which has important implications for the whole body and will have the utmost clinical impact on critically ill patients [45,46,47,48,49].

An important goal during surgery and general anesthesia is maintaining an adequate cerebral blood flow that will sustain normal cerebral metabolic activity and sufficient brain oxygenation. The literature has shown a 2–10% rate of severe cerebral ischemia cases during cardiovascular surgery and 0.05–7% during non-cardiac surgery [50,51,52]. Biochemically, important amounts of ROS and RNS are produced during ischemia–reperfusion. Moreover, an important increase in the intracellular Ca^2+^ levels has also been demonstrated, as well as an exacerbation of the neurotoxic properties of glutamate. Wang et al. studied the molecular effects induced by curcumin-encapsulated nanoparticles on oxidative stress in cerebral tissue that had undergone ischemia–reperfusion injury. They proved a decrease in the oxidative attachment on the endothelial tissue, as well as a decrease in the blood–brain barrier (BBB) [53,54,55].

Recent studies have discussed different theories regarding cerebral metabolism and the regional cerebral blood flow (eCBF) [56,57]. Maintaining an optimal cerebral blood flow is crucial for ensuring continuous global or regional oxygenation. Weiss et al., in an experimental study regarding cerebral oxygen consumption during ischemia and reperfusion, identified a direct correlation between O_2_ supply/consumption and ischemia–reperfusion syndrome. The authors used two study groups of laboratory animals, one group (*n* = 9) where cerebral ischemia was induced and one group (*n* = 9) where reperfusion was induced. The study animals received 90% isoflurane anesthesia with a mixture of oxygen and gas through an ETT tube, with the endpoint of the partial pressure of oxygen of over 100 mmHg. To achieve cerebral ischemia, the authors blocked the medial cerebral artery for 1 h in the case of group 1, while for group 2, they reperfused the same artery for 2 h. Cerebral blood flow was determined using the C^14^-iodoantipyrine autoradiographic technique [58].

In a study on cerebral protection bestowed by anesthetic agents, Sakai et al. reported improved neurological function after focal ischemia. During the study, the researchers induced cerebral ischemia in lab animals through the temporary occlusion of the medial cerebral artery. They proved a reduced incidence of neurological deficits in the animals that suffered from cerebral ischemia under general anesthesia with isoflurane, compared to the control group. Moreover, the authors showed that the cerebral protection induced by isoflurane persists up to eight weeks after ischemia [59].

Another important oxidative mechanism in the case of IR injury is based on the iNOS activity and on increased NO concentrations. The redox implications of NO are interesting as they have an increased reactivity toward other molecules. A specificity of this pathway is the formation of peroxynitrite free radicals following the interaction between NO and the superoxide radical [60,61]. At the cellular level, peroxynitrite attacks the mitochondrial membrane and DNA oxidation, blocking DNA repair mechanisms and leading to cell energy failure [62,63].

Neuronal OS occurring during cerebral ischemia is divided into three major groups of nitric oxide synthases: end endothelial NOS (eNOS), neuronal NOS (nNOS), and inducible NOS (iNOS) [64,65]. Recent studies have shown an increase in the activity of these enzymes during cerebral ischemia, with a significant increase in NO production. In a study of the expression of NO in patients suffering from Moyamoya disease who had undergone bypass surgery with a temporary occlusion of the M4 branch of the middle cerebral artery, Silver et al. showed increased expression of mixed venous nitride (NO_2_) immediately after the occlusion of the cerebral artery. The authors concluded that NO_2_ expression is directly proportional to the degree of cerebral ischemia [66].

During cerebral ischemia, the released OS and the increased flow of reactive species can influence the BBB, leading to a migration of T lymphocytes, macrophages, natural killer cells, and polymorphonuclear leucocytes. Among the most widely studied reactive species involved in the BBB destruction are transforming growth factor beta (TGF-β), interleukin 6 (IL-6), interleukin 10 (IL-10), interleukin 1-beta (IL-1β), interferon beta (IFN-β), and tumor necrosis factor alpha (TNF-α) [50,67,68,69,70]. Boutin et al. proved the destructive impact of Il-1 in ischemic brain injury. Numerous mechanisms correlate increased IL-1β production with an increase in BBB permeability [71].

The most representative chemokines involved in the molecular mechanisms of IR are the monocyte chemoattractant protein-1 (MCP-1/CCL2). The main activity of MCP-1 is the central recruitment of activated lymphocytes, macrophages, and monocytes. Hartley et al. studied the implications of CCL2 in the changes that occur at the BBB level after ischemia. Blocking CCL2 with an antibody significantly reduces the permeability of the BBB [72], together with the distribution of the tight-junction proteins. A similar study carried out by Stamatovic et al. reported that MCP-1 plays an essential role in leukocyte recruitment and in the increase in BBB permeability [73].

One of the main aspects of general anesthesia is mechanical ventilation, which has significant implications for OS expression in the surgical patient. The most common trigger for OS expression at cellular and molecular levels is lung ischemia–reperfusion injury (LIRI). By reducing or even interrupting pulmonary blood flow, the reactive oxygen species production in the endothelium will be accelerated. Another site for excessive ROS production is the alveolar macrophage. The reperfusion phase is characterized by pro-inflammatory cytokine release, the activation of NOS, and neutrophile recruitment [73,74,75,76,77,78] (Figure 1).

Changes in the redox balance represent another frequent phenomenon in the case of LIRI, as an excessive production of reactive oxygen species (ROS) occurs right after the reperfusion. In the lung, there are a series of potential ROS generators, such as activated xanthine oxidase, neutrophiles, mitochondria, the NADPH oxidase system, and NO synthase. In a study, Chen et al. showed that mitochondrial damage plays a more important role in the ischemia process than in the reperfusion process. The oxidative phenomenon is augmented by reduced oxygen delivery in the mitochondria, which leads to a decreased oxidative phosphorylation process and ATP production [80]. Decreased ATP production will lead to a dramatic decrease in the mitochondrial membrane potential, leading to calcium transfer outside the mitochondria and the initiation of cellular apoptosis. Damage to the mitochondrial membrane and permeabilization of the membrane will lead to an increased destruction of certain proteins that are vital for the normal functioning of the mitochondrial system, such as cytochrome-C. Furthermore, the release of cytochrome-C in the cytosol will lead to the activation of caspase-9 and an increase in the apoptotic process at the cellular level [81,82]. NOS found in the lung can be subdivided into four species: endothelial NOS (eNOS), inducible NOS (iNOS), neuronal NOS (nNOS), and mtNOS. Studies have shown an increased pro-inflammatory activity for mtNOS, as well as an increased anti-inflammatory activity for iNOS and eNOS [83,84,85,86].

In their study, Fischer et al. showed that apoptosis is only present after the reperfusion of lung tissue, that the activity increases dramatically, and that the peak is reached after 2 h. Apoptosis is initiated by the activation of both intrinsic and extrinsic mechanisms, with mitochondria as the modulating factor in both pathways [87]. The intrinsic pathway is also known as the mitochondrial pathway and is initiated by the action of cytokines and free radicals. It is characterized by changes in mitochondrial membrane permeability through bcl-2 proteins. The extrinsic pathway, however, is characterized by the activation of certain signaling proteins, such as the TNF receptor, angiotensin 2, and the FAS/FAS ligand [88,89].

Ischemia in the lung tissue activates caspase-8 through the FAS/FAS-ligand system. The next phase is represented by the activation of caspase-7, caspase-6, and caspase-3, leading to DNA fragmentation through the poly-ADP ribose polymerase [88]. During ischemia–reperfusion syndrome, research has shown a rapid release of vasoconstrictor and vasodilatory mediators inside the lung, all derived from arachidonic acid: C3 and C5a complement fragments; thrombin; transcription factor; and a series of pro- and anti-inflammatory cytokines, such as IL-1β, interleukin-2 (IL-2), IL-6, IL-8, interleukin-9 (IL-9), IL-10, IL-18, interferon-γ (INF-γ), and TNF-α.

The consequences of various anesthetic drugs for immune cells have been extensively investigated in in vitro studies. Cultures of immune cells, such as neutrophils, lymphocytes, and/or NK-cells, are separated and exposed to clinical concentrations of anesthetics. Simultaneously, the cells are isolated from the whole network and interaction with other immune cells or mediators that may not reflect the clinical situation.

Several studies have shown that propofol has immuno-inhibitory activity through actions on the non-specific immune system, impairing monocyte and neutrophil functions such as chemotaxis, phagocytosis, and oxidative burst. Some studies have suggested that this activity is due to the lipid solvents found in the formula. The chemotactic and phagocytic function of neutrophils may be inhibited by a decrease in intracellular calcium. The effects of propofol seem to be dose-dependent and occur at clinically relevant concentrations [90,91]. Propofol has been found to suppress the functions of polymorphonuclears, lymphocyte proliferation, and cytokine release in response to endotoxemia only in patients already immunocompromised, but not in healthy volunteers. Hoff et al. investigated the effects of propofol and ketamine on TNF-α gene expression in peripheral blood mononuclear cells and found that lipopolysaccharide endotoxin stimulated TNF-α and that mRNA steady-state transcripts were significantly increased in the presence of propofol (+42%) and decreased in the presence of ketamine (−31%) compared to control cell groups [92].

The effects of most anesthetics are at least partially mediated through GABA-A receptors. Several studies have demonstrated that these receptors can be found on the immune cell membranes and can be influenced by anesthetic drugs. GABA can activate ion channels on macrophages and monocytes and decrease cytokine secretion and T-cell proliferation. Moreover, drugs that influence GABA concentration, such as vigabatrin and muscinol, seem to have the same effects. There is extensive crosstalk between the central nervous system and the immune system, and these findings offer a possible explanation as to why chronic propofol administration in intensive care patients increases the risk of infection. On the contrary, monocyte GABA-A receptors are not influenced by diazepam. Therefore, the use of benzodiazepines as sedative agents may be more appropriate where infection is a life-threatening problem.

The effects of inhalational anesthetics are also mainly inhibitory, for example, suppressed neutrophil function, altered lymphocyte proliferation, and decreased cytokine release by mononuclear cells. In contrast to halogenated inhalational anesthetics, which are known to suppress pro-inflammatory cytokines in murine pulmonary cells, volatile anesthetics induce increased gene expression of pro-inflammatory cytokines. Moreover, some studies have shown that the effects of inhalational anesthetics are dose- and time-dependent [93,94,95].

A 2013 study suggested that the NF-κB signaling pathway contributes to sevoflurane- and isoflurane-induced neuroinflammation by increasing the levels of IL-6 (isoflurane by 410% and sevoflurane by 290%) and the nuclear levels of NF-κB (isoflurane by 170% and sevoflurane by 320%). Other studies have confirmed that isoflurane can increase the levels of IL-6, which has been associated with learning and memory impairment in animals [96,97].

Several in vitro studies have reported changes in the immune and inflammatory expression of morphine and other opioids. Among the effects of morphine are altered cytokine release and lymphocyte proliferation, activated enzymes involved in macrophage and lymphocyte apoptosis, inhibition of cell adhesion, and increased secretion of stress hormones. Synthetic opioids, except for fentanyl, seem to have a less pronounced impact on the immune system, probably because they do not interact with the µ2 opioid receptors on immune-competent cells [98,99].

Intravenous rocuronium bromide induces pain during injection and/or withdrawal movement, the exact mechanism of which is not yet understood. One study found that rocuronium bromide induces inflammation in calf cells by inhibiting endothelial nitric oxide synthase, suppressing nitric oxide production, activating cyclo-oxygenase-2, and increasing prostaglandin E2 synthesis, which may be the cause of pain during injection [100].

Challenges in isolating various immune variables from the complex immunological network in clinical scenarios lead to far fewer in vivo than in vitro studies. Due to their volatile nature, inhalational agents are important not only to the patients but also to the personnel attending the operating theater. Chronic occupational exposure to low doses of isoflurane were correlated with fewer total T and T-helper (CD4^+^) lymphocytes in a dose- and time-dependent manner [101]. The in vivo effects of sevoflurane and isoflurane largely reflect the in vitro studies. Several studies have demonstrated that these two agents can reduce the levels of reactive species of oxygen, inhibit chemotaxis, and inhibit the activation of neutrophils, leading to decreased neutrophil adhesion to human endothelial cells, decreased IL-6 and TNF-α, suppressed PMN migration and function through various pathways, altered cytokine release from macrophages, and decreased NK cytotoxic activity. A study comparing two anesthetic techniques, TIVA and balanced inhalational anesthesia, found that perioperative cell-mediated immunity was less influenced by TIVA, suggesting that the stress response in patients undergoing anesthesia with a volatile agent is more pronounced [102,103,104,105,106,107].

A 2019 study on the effects of propofol on neutrophil function, lipid peroxidation, and the inflammatory response during elective coronary artery bypass grafting in patients with impaired ventricular function found that the addition of propofol contributes to a lesser increase in IL-6 levels and a greater decrease in IL-10 compared to placebo. No effects were seen on IL-8. In addition, malondialdehyde (MDA), a marker of lipid peroxidation and free-radical injury, measured in the coronary artery sinus, was lower in the propofol group at 1, 3, 5, and 60 min after reperfusion (*p* < 0.01), suggesting that propofol attenuates free-radical-mediated lipid peroxidation and systemic inflammation in patients with impaired myocardial function undergoing CABG [108].

Potocnik et al. studied patients undergoing thoracic surgery and one-lung ventilation where either propofol or sevoflurane was used to maintain the anesthesia. They found that IL-6, IL-8, and CRP levels were higher in the propofol group, while IL-10 was higher in the sevoflurane group. The oxygenation index 6 h after the surgery was lower in the propofol group compared to the sevoflurane group (*p* = 0.02), maintained 24 h after the surgery (*p* = 0.019). The number of postoperative adverse events was significantly higher in the propofol group (*p* < 0.05) and included ARDS, pneumonia, and SIRS [109]. In a similar manner, but on patients undergoing craniotomy, propofol proved to be less inflammatory than sevoflurane, with a lower IL-6/IL-10 concentration ratio during and at the end of surgery (*p* = 0.0001) and higher IL-10 [110].

A 2019 study compared the effects of propofol with those of desflurane on inflammation and ischemia–reperfusion syndrome during robot-assisted laparoscopic radical prostatectomy (RALRP) and found that propofol-based anesthesia significantly attenuated the increase in IL-6 levels during RALRP (4.68 ± 2.76 pg/mL vs. 8.57 ± 3.72 pg/mL; *p* < 0.001), but with similar effects on TNF-α, CRP, and NO levels [111].

The American Heart Association (AHA) recommends that patients at risk of myocardial ischemia during surgery who are hemodynamically stable should undergo general anesthesia with inhalational agents [112].

These recommendations and the benefits of these agents have been emphasized in numerous studies. A study conducted on 74 patients scheduled for coronary artery bypass graft surgery under cardioplegic arrest showed that 10 min of preconditioning with sevoflurane decreased the release of brain natriuretic peptide in the perioperative period. Moreover, the levels of plasma cystatin C were lower in sevoflurane-preconditioned patients, suggesting improvement in both cardiac and renal function after major heart surgery [113].

Fukazawa et al. presented extensive evidence, both in vitro and in vivo, that modern inhalational halogenated anesthetics play an important role in the pathophysiology leading to AKI by inhibiting renal tubular and endothelial cell necrosis and apoptosis, and inducing anti-inflammatory effects at this level by activating pathways responsible for anti-inflammatory and cytoprotective signaling molecules, such as releasing TGF-β1, activating CD73, inducing IL-11, and generating adenosine [114]. A systematic review concluded that patients undergoing cardiac surgery had a significantly lower increase in creatinine levels in the first two days after surgery and lower intensive care unit stay and hospitalization when anesthetized with volatile anesthetics compared to TIVA propofol. However, there was no statistically significant difference in mortality between the groups [115]. Furthermore, current anesthetics seem to offer neuroprotection, sustained for 2–4 weeks, against mild or moderate ischemic injury but have no influence on severe ischemia. These effects are mediated by antagonizing postsynaptic glutamate receptors and enhancing GABA-A-mediated hyperpolarization and by inhibiting several processes that lead to neural apoptosis (increased levels of antiapoptotic proteins, such as bcl-2 and reduced cytochrome-C release) [116,117,118].

The benefits of inhalational anesthetics extend beyond the operating theater, with extensive research suggesting that sedation with sevoflurane or isoflurane in the intensive care unit plays an important role in preventing ventilatory-associated lung injury (VILI). The release of pro-inflammatory markers, such as IL-1b and MIP-1β, and reactive oxygen species, as well as neutrophil transmigration into the alveolar compartments of the lungs are rapidly prevented in the early stages of VILI if sedation is switched from intravenous to inhalational [119]. Moreover, inhalational anesthetics seem to influence glycemic control in the perioperative period. Blood glucose levels were found to be higher with volatile agents, such as sevoflurane and isoflurane, compared to propofol-maintained anesthesia. This response was attributed to impaired glucose clearance and increased glucose production and inhibition of normal insulin production.

Kalimeris et al. studied the antioxidant effects of propofol on patients undergoing carotid endarterectomy. The study protocol included a control group, with 21 patients who were administered general anesthesia (GA) using sevoflurane, and a study group, in which GA was administered using propofol (*n* = 23). The biomarkers used for the quantification of oxidative stress included the expression of P-selectin and S100B protein, the lactate levels, nitrate, nitrite, and malondialdehyde. The study showed statistically significant differences regarding S100B levels that were lower in the propofol group. Moreover, it showed lower levels of malondialdehyde (MDA), nitrates, and nitrites in the same propofol group [120]. Yang et al. carried out a similar study regarding the inflammatory influence of sevoflurane vs. isoflurane in an experimental ischemia–reperfusion model. To quantify the impact on OS, they analyzed certain biomarkers, such as superoxide dismutase (SOD), MDA, nitric oxide (NO), and complement C3, C5a, and C6. The study reported lower expressions of OS in the sevoflurane group [121] (Table 1).

## 3. Implications of Genetic and Epigenetic Expression on Inflammation and Oxidative Pathways

microRNA species are biosynthesized in the cell nucleus through the action of RNA polymerase II on specific genes. The initial species that result after the first interaction are called pri-microRNAs [152]. The chain of biochemical processes continues with the attack of RNA polymerase III endonuclease (Drosha) on the pri-microRNAs, leading to the formation of pre-microRNAs. This reaction is catalyzed by DiGeorge Syndrome Critical Region 8 (DGCR8) [153]. The biosynthesis process continues in the cell cytoplasm, where pre-microRNAs are transported from the nucleus to the cytoplasm with the help of Exportin-5. The final microRNA species result from a reaction between the pre-microRNAs and Enase III endonuclease (Dicer) and the RNA binding protein (TRBP). The newly formed epigenetic species are then transported inside the cell as microvesicles, exosomes, ribonucleoprotein complexes, and high-density lipoproteins [154]. Recent studies have shown that anesthetic drugs induce a series of molecular changes that are especially involved in the modulation of the inflammatory and oxidative chains. Kim et al. studied the process and identified a decrease in the expression of microRNA-27a, microRNA-194, microRNA-23b, microRNA-133a, microRNA-199a, microRNA 101, microRNA-219-5p, microRNA-30b, microRNA-92b, microRNA-127, and microRNA-204. In the same study, they reported increases in the expression of microRNA-370, microRNA-134, microRNA-302a, microRNA-107, microRNA-92a, microRNA-202, microRNA-190b, microRNA-29b, microRNA-216a, microRNA-181d, and microRNA208b [155]. A similar study was carried out by Twaroski et al. and showed essential changes in the expression of miRNA-21 in patients who had received propofol [156].

The brain is one of the most sensitive organs that must be protected during GA. Anesthetic management and perioperative protocols especially focus on maintaining adequate cerebral perfusion and oxygen delivery. The pathophysiological mechanisms involved in maintaining a constant and adequate cerebral perfusion are complex and not fully understood. It is, however, a known fact that cerebral autoregulation maintains a constant cerebral blood flow despite the changes in mean arterial pressure. Biological functions are extremely sensitive, even to the slightest change in mean arterial pressure, and therefore the cerebral autoregulation mechanisms will be activated to prevent cellular anoxia [157]. From an epigenetic point of view, scientists have identified different involvements of microRNAs in the biological pathways related to cerebral ischemia and reperfusion. In a study regarding the expression of microRNAs in cerebral ischemia in experimental animals, Gusar et al. identified statistically significant changes up to 48 h after cerebral ischemia for a series of epigenetic species, such as microRNA-30c-2-3p, microRNA-125b-5p, microRNA-300-5p, microRNA-107-3p, microRNA-497-5p, microRNA-27a-3p, microRNA-30b-3p, microRNA-29c-3p, microRNA-124-3p, microRNA-330-5p, microRNA-181b-5p, microRNA-29a-3p, let-7i-3p, microRNA-22-3p, let-7i-5p, microRNA-212-3p, microRNA-26a-5p, microRNA-99a-3p, microRNA-221-3p, microRNA-27b-3p, microRNA-24-3p, microRNA-22a-5p, microRNA-29a-5p, microRNA-186-5p, microRNA-99a-5p, microRNA-128-3p, microRNA-23a-3p, microRNA-376b-5p, microRNA-30c-5p, microRNA-9a-5p, microRNA-30a-3p, let-7f-5p, microRNA-21-5p, and microRNA-223-3p [158]. A molecular mechanism associated with cell survivor apoptosis is phosphatase and tensin homologous protein (PTEN) and its involvement in the biochemical activity of PI3K/AKT [159]. In a study on the cellular and epigenetic mechanisms involved in initiating, augmenting, and maintaining cerebral ischemic injury, Wang et al. showed that microRNA-32-5p activity is directly proportional to the expression of PTEN/P13K/AKT, which is responsible for protecting cerebral tissue. Huang et al. performed a similar study to present the role of microRNA-326-5P in reducing the signal transducer and activator signal of transcription-3 (STAT3) and the protection of cerebral tissue during ischemia–reperfusion syndrome [160].

Other mechanisms involved in cerebral injury following ischemia and reperfusion are the excessive production of free radicals, mitochondrial disfunction, lipid degeneration, protein oxidation, DNA structure modification, and finally cerebral tissue death. In an experimental study, Guo et al. reported a series of neuronal protection mechanisms through the activity of microRNA-25 [161]. Huang et al. described that ischemia and reperfusion lead to augmented tissue damage and accelerated cell death [162]. In a complex study on the actions of microRNA-25 in neuronal protection during cerebral ischemia, Zang et al. reported that the increased expression of microRNA-25 inhibits Fas/FasL activity and reduces apoptosis induced by ischemic injury [163]. Mao et al. reported protective effects at the cerebral level when p38 mitogen-activated protein kinase (MAPK) activity is inhibited through microRNA-128-3p. Moreover, they showed that higher levels of microRNA-128-3p are able to block the post-transcription activity of p38α and can contribute to the survival of neurons during cerebral ischemia [164].

An experimental study carried out on laboratory animals by Zhang et al., investigating the expression of certain biological markers, such as microRNAs-144, metastasis-associated lung adenocarcinoma transcript 1 (MALAT1), and glycogen synthase kinase-3β (GSK3β) in LIRI, showed that the overexpression of genes was directly proportional to the intensity of the ischemic injury. Regarding the epigenetic expression of microRNA-144, the authors reported a decrease in its concentration [165]. Li et al. studied the expression of microRNA-146a in LIRI and showed a decrease in the epigenetic expression in pulmonary ischemic injury. The group identified that microRNA-146a activity is directly proportional to the overproduction of IL-1β, IFN-γ, TGF-β1, and TNF-α [166]. Zhou et al. reported an increase in the expression of microRNA-155 during LIRI, accompanied by an overexpression of IL-6, IL-1β, TNF-α, 8-isoprostaglandin F2α (8-iso PGD2α), and 8-hydroxy-2-deoxyguanosine (8-OHdG) [167]. Overexpressed in LIRI, microRNA-223 is involved in the hypoxia-inducible factor-2α (HIF2α) pathway, being responsible for worsening the pro-inflammatory status in the lung [168]. Cai et al. carried out a study on the expression of microRNAs in pulmonary ischemia in laboratory animals that had undergone pulmonary transplantation, discovering increased microRNA-206 levels [169]. Another experimental study reported decreased levels of microRNA-18a-5p in ischemic pulmonary tissue [170]. Qin et al. studied the expression of microRNAs in multiorgan ischemia caused by hemorrhagic shock and identified an increased expression of microRNA-24a in the animal group in which hemorrhagic shock had been induced (*p* < 0.05). Qin et al. also identified a statistically positive correlation between increased microRNA-34a levels and serum malondialdehyde (MDA) [171]. Li et al. showed that the expression of microRNA-21-5p and IL-1β, IL-6, IL-8, IL-17, and TNF-α bioproduction by macrophages are statistically dependent in the case of laboratory animals suffering from pulmonary ischemia [172].

During general anesthesia, one of the main pathophysiological factors the anesthetic team needs to consider is the maintenance of blood-pressure homeostasis in order to ensure the adequate oxygenation of vital organs. An important marker for the adequate perfusion of tissues is provided by kidney function, assessed through the excretion capacity. In many clinical scenarios, the kidneys suffer both due to tissue hypoperfusion and due to other factors, such as inflammation, rhabdomyolysis, and hemorrhagic shock. One of the main causes of acute kidney injury is ischemia–reperfusion syndrome. From a pathophysiological standpoint, kidney ischemia–reperfusion syndrome is complex and multifactorial, with not all causes being fully elucidated at this point. Among the basic causes of kidney IR injury is the destruction of renal tubular cells, leading to the destruction of renal tubules. Clinically, once these pathophysiological mechanisms have exerted their effects, the glomerular filtration rate (GFR) has already been affected. Dixon et al. identified a specific mechanism for the destruction of renal cells, characterized by lipid peroxidation, condensation of mitochondrial membranes, and iron accumulation. The process, named ferroptosis, has been recently identified and explained in other studies as being involved in different pathological states, including myocardial infarction [173,174,175]. Inside the cell, there are a series of enzymes and proteins responsible for the inhibition of ferroptosis, such as glutathione peroxidase 4 (GPX4), responsible for diminishing the effects of lipid peroxidation, a fact proven by Yang et al. in a study regarding ferroptosis mechanisms in cancer cells [176]. Fang et al. studied another mechanism possibly responsible for ferroptosis inhibition, the reduced activity of lipid peroxidation through the transport of cystine in the cytosol and through the stimulation of GSH production through Solute carrier family 7 member 11 (SLC7A11). Another molecular and epigenetic mechanism involved in increasing renal disfunction is represented by microRNAs [177]. According to Drobna et al., it is possible to augment molecular mechanisms through the action of genes and microRNA-20b-5p and microRNA 363-3p [178]. In a similar study, Ding et al. proved the direct involvement of microRNA-182-5p and microRNA-378a-3p in intensifying the ferroptosis process and renal function deterioration due to ischemia and reperfusion [179].

In an experimental study on the role of microRNAs in kidney ischemia, Chiba et al. indicated that the subjects presented high levels of microRNA-17, microRNA-18a, microRNA-19a, microRNA-20a, microRNA19b-1, and microRNA-92a-1 [180]. MicroRNA-24 was identified as a promoter of ischemic kidney injury, acting by stimulating the biological mechanisms of H2A-histone family and oxygenase 1 [181]. Wang et al. also discovered increased expressions of microRNA-466 and microRNA-709 in kidney disfunction induced by ischemia [182]. Furthermore, this group of authors reported the important involvement of genes Mcm5, Oip5, E2f8, Myc, E2f1, and Mdm2 in increasing the severity of ischemic injury. A similar study was carried out by Cao et al., and it identified significant implications of microRNA-125b-5p in ischemic kidney injury [183]. MicroRNA-489 was found to be a protective epigenetic species against kidney disfunction induced by ischemia through hypoxia-inducible factor-1 [184]. Other studies have found a series of microRNA species whose expression was significantly modified by acute renal injury, such as microRNA-4640, microRNA-4270, microRNA-4321, microRNA-10a-5p, microRNA-26b-5p, microRNA-16, microRNA-27a-3p, microRNA-93-3p, microRNA-29a-3p, microRNA-126-3p, microRNA-127-3p, microRNA-200c, microRNA-210-3p, microRNA-210, microRNA-423, microRNA-21, microRNA-30e, and microRNA-30a [181,185,186,187,188,189,190,191]. In a study on microRNA expression in patients presenting with septic shock and acute kidney injury (AKI), Zang et al. identified decreased plasma and urine levels for microRNA-22-3p in patients who did not survive. The statistical analysis based on the Kaplan–Maier survival curves and Cox regression analysis has identified a statistically significant correlation between a decrease in this microRNA species and 28-day mortality for patients presenting with septic shock and AKI [192]. In the case of critically ill patients, one of the most widely studied mechanisms of kidney injury is the formation of oxalic acid and calcium oxalate (CaC_2_O_4_), which produce tubular obstruction and crystallization in epithelial cells. Shihana et al. identified increased expressions of microRNA-20a, microRNA-93, microRNA-92a, microRNA-195, and microRNA-451 in acute kidney injury caused by oxalate [193]. Another mechanism responsible for worsening kidney tissue injury is the activation of mitogen-activated protein kinase ½ (MEK1/2) through phosphorylation. Recent studies have reported the implications of MEK1/2 in differentiation, migration, proliferation, metabolism, and cellular death. Collier et al. studied all these complex mechanisms and reported the active involvement of microRNA-34a in decreasing MEK1/2 activity and initiating ischemic renal injury [194]. Another microRNA species involved in pathogenetic mechanisms is microRNA-21 [195]. Zhang et al. showed that microRNA-192 expression increases in patients with AKI. Moreover, there is an increase in the concentration at 3 h after the ischemic event in the kidney, with a peak value at 12 h, and a decrease in its levels during the next 24 h (AUC-ROC 0.673, 95% CI 0.540–0.806, and *p* = 0.014) [196]. The Wnt/β-catenin pathway is another mechanism involved in kidney ischemia and is responsible for fibrosis in the kidney [197,198]. Apart from the aforementioned microRNAs, there is a molecular complex responsible for modulating the activity of Wnt/β-catenin, which includes Dickkopf (DKK) 1, 2, 3, and 4 (DKK1, DKK2, DKK3, and DKK4) [199]. Yi et al. showed that microRNA-214 is directly involved in inhibiting the activity of Wnt/β by directly blocking β-catenin [200]. Zu et al. identified the direct molecular processes of microRNA-214 in kidney injury through the modulation of Wnt/β-catenin activity and the expression of DKK3 [201].

## 4. Perioperative Antioxidant Therapy

Numerous studies in the preceding decades have focused on reducing the pro-oxidative profile in surgical patients [202]. Different antioxidant therapies have been proposed, targeting either the inhibition of biological processes responsible for the production of certain molecules or the neutralization of the redox activity of free radicals by complexing them in molecular formulas with no chemical activity [203,204,205]. One such widely studied molecule is hydrogen sulfide (H_2_S), which due to its structural properties can easily diffuse through cell membranes and through the membranes of organelles. From a biochemical standpoint, the antioxidant activity of H_2_S is represented by the inhibition of reactive oxygen species through nuclear factor erythroid 2-related factor 2 (Nrf2) activation and Kelch-like ECH-associated protein-1 (Keap1) inactivation and by an increase in GSH and Trx expression. The main sources of H_2_S biosynthesis in the body are cystathione β-synthase (CBS), predominantly found in the central nervous system (CNS), and cystathionine γ-lyase (CSE), found in the cardiovascular system (CVS). The biosynthesis mechanism is based on the desulfydration of cysteine through catalyst biological mechanisms given by the pyridoxal-5′-phosphate-dependent enzymes. Tripatara et al. carried out an experimental study on 74 lab animals (Wistar rats) on the effect of endogen and exogen H_2_S on the ischemia–reperfusion syndrome in the kidney tissue. Exogen H_2_S sources were represented by the administered sodium hydrosulfide, which reduced the incidence and severity of tubular and glomerular dysfunction in the experimental model that had undergone 45 min of ischemia and 6 h of reperfusion. A histochemical analysis showed a significant reduction in the histological score for acute tubular necrosis and, importantly, a decrease in the expression of redox mechanisms [206]. Bos et al., in a similar study, reported a protective effect of H_2_S for renal function, apoptosis, and kidney tissue inflammation under experimental conditions of ischemia–reperfusion injury. They highlighted that by inducing a hypermetabolic status through H_2_S, the kidney can be protected from hypoxia [207]. The most recent studies have focused on blocking biochemical mechanisms of free radical production inside the mitochondria. In this regard, one of the most widely studied pharmacological formulas is represented by SS-31, also known as Bendavia, Elamipretide, or MTP-131. Structurally, it is a synthetic tetrapeptide with the formula D-Arg-2′6′dimethylTyr-Lys-Phe-NH_2_ [208]. SS-31 can selectively attack cardiolipin from inside the mitochondrial membrane and is able to block the transient mitochondrial permeability mechanism that is in action during ischemia–reperfusion, leading to increased ATP biosynthesis and reduced ischemic injury [209]. Szeto et al. identified low levels of IL-18, IL-1β, and TGF-β in ischemic tissue. They reported a decrease in macrophage infiltration and TNF-α activity, with the restoration of glomerular capillaries and the podocyte structure [210]. Birk et al. carried out a similar study and reported the recovery of the ATP production mechanism in the mitochondria during kidney ischemia–reperfusion syndrome [209]. In addition, 4-hydroxy-2,2,6,6-tetramethylpiperidine-N-oxyl, also known as Tempol, was investigated for reducing cellular redox activity, due to its capacity to block reactive oxygen species by reducing intracellular Fe^2+^ concentrations [211]. Chatterjee et al., in an experimental study on Wistar rats on the effects of Tempor on redox activity in the proximal renal tubules, showed that there was a significant decrease in renal disfunction and kidney injury due to ischemia–reperfusion syndrome. The mechanisms of action that were identified in this case are characterized by a reduction in hydrogen peroxide biosynthesis in the respiratory chain and an increase in lactate dehydrogenase (LDH) [212].

Patients under surgical stress undergo a series of time-related changes at a metabolic level. More et al., in their book published in 1959, described the stages of surgical stress as follows: Phase 1—stress period that can continue for 2–4 days; Phase II—turning point, which will be followed by the anabolic phase; Phase III—regeneration of proteins and muscle, which happens a few weeks after surgery; and Phase IV—accumulation of fat [213]. During these phases, muscular proteins are broken down to activate gluconeogenesis, energy production, and tissue repair, some of the reasons for which the nutrition of the surgical patients is of the utmost importance before and after surgery [214,215]. Miyachi et al., in a study on the impact of cysteine administration (700 mg) and theanine (280 mg) for four days before surgery and five days after surgery, in oncological patients undergoing distal gastrectomy, showed reduced postoperative inflammation and faster recovery [216]. A similar experimental study carried out by Shibakusa et al. reported a decrease in IL-6 expression in laboratory animals that had received cystine/theanine five days before surgery [217]. Tanaka et al. came to similar conclusions, stating that cystine administration leads to a decrease in the concentration of IL-6 and IL-10 in lab animals with induced sepsis [218].

The most widely studied molecule with antioxidant properties is vitamin C. From a chemical perspective, ascorbic acid (AscH_2_) is a water-soluble ketolactone that expresses two ionizable hydroxyl groups (Sven-Olaf Kuhn). At physiological pH, the predominant chemical type is the AscH^−^ anion. The redox capacity of ascorbate is given by its reduction in two steps [219,220,221,222]: (i) the donation of an electron and (ii) the formation of the ascorbate radical (Asc^−^) and of dehydroascorbic acid (DHA) [223]. Physiologically, the concentration of ascorbate in plasma in healthy individuals is between 40 and 80 microM, a level capable of ensuring endogenous antioxidant activity. The main reactive species to which ascorbate donates an electron leading to a blockage in redox activity are the alkoxyl radical (RO), thiol radical (GS), hydroxyl radical (HO), peroxyl radical (LOO), and tocopheroxyl radical (TO) [224,225,226]. Recent studies have demonstrated that ascorbate can inhibit lipid oxidation processes in endothelial cells and macrophages [227]. Du et al. showed in their experimental study the ability of vitamin C to inhibit inflammatory and pro-oxidative processes mediated by NF-κB [228]. Brown et al. performed a similar study regarding the impact that vitamin E and vitamin C antioxidant therapy has on pancreatic transplantation and showed a significant decrease in the inflammatory and oxidative expressions [229]. Moon et al. carried out a prospective, randomized study that included 132 patients scheduled for gynecological surgery. For statistical analysis and for underlining the antioxidant effects of vitamin C, the authors divided the patients into four groups as follows: Group 1 (*n* = 33), which received magnesium sulfate 40 mg/kg; Group 2 (*n* = 33), which received vitamin C 50 mg/kg; Group 3, which received magnesium sulfate 40 mg/kg and vitamin C 50 mg/kg; and Group 4, which received isotonic saline solution 40 mL. The monitored clinical effects were the postoperative pain score, postoperative nausea and vomiting (PONV), and total fentanyl requirements. The study showed significantly lower fentanyl requirements in the groups that received magnesium sulfate and vitamin C (Group 1, Group 2, and Group 3). The incidence of PONV was lower in the group that received both magnesium sulfate and vitamin C [230]. Mohamed et al. studied the impact of perioperative intravenous administration of vitamin C and N-acetylcysteine on the production of malondialdehyde, IL-6, and IL-8 as markers for oxidative stress and systemic inflammation in patients needing a tourniquet during surgery; the study included 60 patients who had undergone surgical intervention of the upper limb and received 1 g of vitamin C and 10 mg/kg N-acetyl cysteine. Following statistical analysis, there was an increased expression of IL-6, IL-8, and malondialdehyde in patients who had not received antioxidant substances [231]. Rumelin et al. studied the plasmatic concentration of vitamin C in the pre- and postoperative period following the administration of 1 g of single-dose vitamin C. For the study, the authors divided the patients into two groups: one group that received 1000 mg vitamin C preoperatively and one group that received a placebo. The vitamin C levels in plasma were determined in both groups on the first postoperative day. The study concluded that the administration of 1 g of vitamin C in the preoperative period does not ensure a constant increased plasma concentration [232]. Jeon et al. carried out a similar study with 100 patients undergoing laparoscopic colectomy who received 50 mg/kg intravenous vitamin C immediately after the induction of anesthesia. The authors wanted to monitor the postoperative morphine requirements, as well as the pain scores, at 2, 4, and 24 h after surgery. They identified a decrease in morphine requirements 2 h after surgery in patients who had received antioxidant therapy, as well as a lower pain score at 24 h [233].

Recently, the antioxidant vitamin D has been intensely studied, especially due to the inflammatory mechanisms that are induced upon infection by the new SARS-CoV-2 coronavirus [234,235]. Recent studies have shown that decreased plasma concentrations for vitamin D are associated with an increase in pro-inflammatory and pro-oxidative factors, such as TNF-α, IL-6, fibrinogen, and C reactive protein [236]. It was shown that the vascular and microvascular systems contain a number of important vitamin D receptors (vitamin D), explaining its ability to modulate pro-inflammatory mechanisms at the vascular level. Nieuwland et al., in an experimental study regarding the impact of vitamin D on vascular inflammation, identified a decrease in CD4^+^ T-helper cell count expression. A reduction in the biosynthesis of IL-2, IL-4, and IL-10 associated with T-cell activity was also proven [237]. Martinez-Moreno et al., in a similar study, identified a decrease in pro-inflammatory cytokines, such as IL-6, IL-8, IL-1β, and TNF-α, when human aortic smooth muscle cells are exposed to increased concentrations of vitamin D [238]. The importance of vitamin D was also noticed in the case of orthopedic surgery. Bogunovic et al. carried out a retrospective study that included 723 patients who had undergone orthopedic surgery. In these patients, the levels of 25-hydroxyvitamin D were determined in the preoperative period. The prevalence for vitamin D plasma concentrations was expressed as follows: normal (≥32 ng/mL), insufficient (<32 ng/mL), and deficit (<20 ng/mL). It was shown that 40% of the patients had vitamin D levels <32 ng/mL, and out of these patients, half had levels <20 ng/mL. The authors found a statistically significant correlation between vitamin D levels in plasma and bone fractures or degeneration, underlining the importance of this vitamin in maintaining normal muscle and skeletal functions [239]. Maniar et al. studied the impact vitamin D has on recovery after total knee arthroplasty. The study included 120 patients, out of which 64 had vitamin D levels <30 ng/mL preoperatively. In the postoperative period, all patients received vitamin D antioxidant therapy. Patients with a vitamin D deficit before surgery presented slower recovery in comparison to the others. Analysis performed three months after surgery showed that the functional scores were similar in the two groups [240]. Mouli et al. carried out a similar study, including 174 patients with vitamin D deficit (<30 ng/mL) who received vitamin D antioxidant therapy with the following protocol: (1) daily vitamin D supplementation with an increasing dose between 1000 and 6000 IU and (2) one weekly dose of 50,000 IU, administered for four weeks, followed by the maintenance of a daily dose of 2000 IU. In the surgical patients included in the study, the authors recommended a vitamin D antioxidant therapy with 50,000 IU of vitamin D weekly for four weeks, followed by a maintenance dose, as a better solution than the administration of lower, daily doses [241]. Hajimohammadebrahim-Ketabforoush et al. studied the impact of vitamin D administration in deficient patients that had undergone craniectomy for brain tumor resection. The study included two patient groups that had lower-than-normal vitamin D levels (<20 ng/mL). One of the groups received intramuscular vitamin D 300.000 IU in the preoperative period. In the study group, the vitamin D concentration increased significantly at five days after surgery (*p* < 0.01). The ICU admission time and the duration of hospital stay for patients who received vitamin D supplementation in the preoperative period were also significantly shorter. The authors of the study concluded that antioxidant therapy with vitamin D in the perioperative period is beneficial in patients with brain tumors [242] (Table 2).

## 5. Conclusions

The pro-inflammatory and pro-oxidative profiles of surgical patients undergoing general anesthesia are similar to those of patients presenting with sepsis or septic shock. One reason is that the pathophysiological mechanisms are similar and mainly represented by an increase in the inflammatory profile, as well as microvascular injury due to ischemia and reperfusion. The epigenetic analysis of the microRNA expression shows a series of molecular and cellular mechanisms responsible for either the initiation or the augmentation of this systemic inflammatory profile. Once identified, these pathways can be blocked and/or certain mechanisms can be minimized in order to reduce the systemic impact on the surgical patient. Moreover, the administration of antioxidant therapies indicates a beneficial clinical impact, the most important benefit being reduced postoperative adverse effects. However, further studies are needed regarding the administration of modern antioxidant therapies targeted at the cellular and molecular levels. 

## Figures and Tables

**Figure 1 cells-11-01880-f001:**
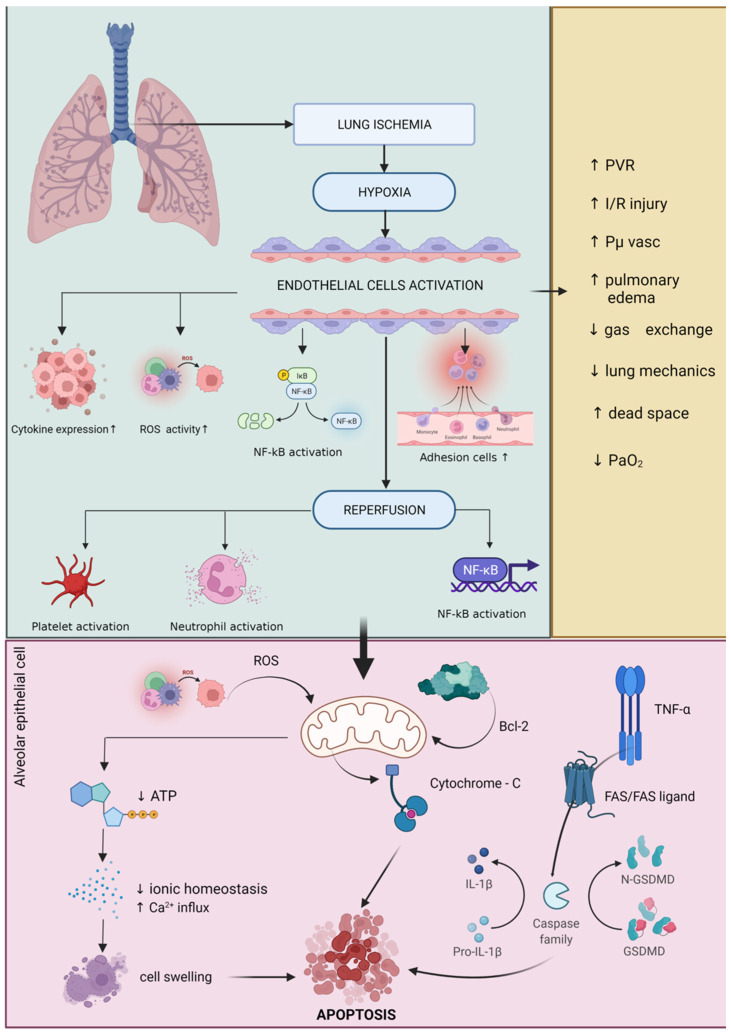
Schematic representation of lung ischemia–reperfusion injury. Green border: Lung ischemia causes a high degree of hypoxia that directly affects endothelial tissue with cytokine activation, increased cell adhesion activity, NF-κB activation, and accelerated generation of reactive oxygen species (ROS). Following reperfusion, a number of other biological mechanisms responsible for increasing pro-inflammatory status (e.g., NF-κB activation), platelet activation, and neutrophil activation. Pink border: these mechanisms directly affect cellular activity by altering mitochondria-specific biochemical pathways. The direct attack of ROS on the mitochondria lowers ATP biosynthesis and affects mitochondrial electrolytic homeostasis, especially by increasing the influx of Ca^2+^ leading to cell swelling and apoptosis. Another biochemical pathway with a significant negative effect on mitochondrial activity is the activation of TNF-α, which activates caspase family via FAS/FAS-ligand and generates increased amounts of IL-1β by catalyzing pro-IL-1β factor. Another mechanism found in cellular apoptosis due to damage to alveolar epithelial cells is the direct action of bcl-2 on the mitochondria leading to cytochrome-C overexpression and finally cellular apoptosis. Yellow border: all these biochemical mechanisms that are involved in the process of mitochondrial and cellular denaturation induced by the phenomenon of ischemia–reperfusion of lung tissue lead to increased pulmonary vascular resistance (PVR), increased pulmonary edema, increased oncotic pressure of the vascular capacity, decreased lung mechanics, increased dead space and decreased adequate oxygenation. Increased expression of reactive oxygen species leads to changes in the cellular and molecular activities in lung tissue, mainly affecting the expression of calcium/calmodulin-dependent NO, nicotinamide adenine dinucleotide phosphate (NADPH), and nuclear factor kappa B (NF-κB). The acceleration of these processes inside the cell leads to pulmonary edema, resulting from an increase in pulmonary vascular resistance and endothelial damage. During mechanical ventilation, these secondary phenomena lead to decreased ventilation and impaired gas exchange in the alveoli. After reperfusion, pulmonary vascular resistance can increase by up to 100%, mainly due to vasoconstriction and endothelial damage in the lung microvasculature. Increased vascular resistance, worsening pulmonary edema, and increased extravascular lung water lead to impaired gas exchange and impaired lung mechanics. All these phenomena negatively impact the clinical status of the patient through a sudden decrease in arterial partial oxygen pressure and an increase in the peak airway pressure and the alveolar–arterial oxygen gradient (A-a/DO2) [78,79]. Recent studies have shown that during pulmonary reperfusion, microvascular permeability can increase by up to 10 times. It was suggested that the initial stage depends on the production of interleukin-8 (IL-8), interleukin 12 (IL-12), interleukin-18 (IL-18), and TNF-α, while the second stage is responsible for the production of activated neutrophiles and pro-oxidative factors, such as interleukin-8 (IL-8), leukocyte adhesion molecule CD18, endothelial P-selectin, and endothelial intracellular adhesion molecule-1 [74].

**Table 1 cells-11-01880-t001:** The impact of anesthetic drugs on inflammatory expression and cellular pathways activity.

Author	Anesthetic Drug	Study Type	Comments	References
Gyires et al.	Morphine	in vivo	↓ edema; ↓ bradykinin activity; no influence on edema induced by histamine;	[122]
Sacerdote et al.	Morphine	in vivo	↓ inflammatory response; ↑ activation of mu-opioid receptors;	[123]
Planas et al.	Liposomal morphine	in vivo	↓ inflammatory edema; ↑duration of effects by encapsulation;	[124]
Joris et al.	Morphine	in vivo	↓ vascular inflammation; ↓ vascular permeability;	[125]
Honore et al.	Morphine	in vivo	↓ edema; ↓ Fos-like immunoreactive activity;	[126]
Jin et al.	Endomorphin-1 (Endogenous ligand for mu opioid receptor)	in vivo	↓ peripheral edema; ↓ activity of Fos-like immunoreactive expression;	[127]
Fletcher et al.	Morphine	in vivo	↓ inflammation; ↓ edema;	[128]
Walker et al.	Kappa-opioid drugs	in vivo	↓ adhesion molecules; ↓ TNF expression; ↓ expression of inflammation cascade;	[129]
Hu et al.	Sufentanil	in vivo	↑ super oxide dismutase (SOD) activity; ↑ catalase (CAT) activity; ↑ glutathione peroxidase (GSH-Px) activity; ↓ malondialdehyde (MDA) expression; ↓ Nuclear factor erythroid 2—related factor 2 (Nrf2) expression; ↓ inflammation and oxidative stress;	[130]
Rahimi et al.	Morphine	in vivo	↓ neuroinflammation via opioid receptors; ↑ neurobehavioral function after traumatic brain injury; ↓ blood brain barrier (BBB) leakage;	[131]
Zhou et al.	Sufentanil	in vivo	↓ Activating transcription factor 4 (AFT4) expression; ↓ liver inflammation; ↓ apoptosis injury induced by ischemia–reperfusion injury;	[132]
Hofbauer et al.	Remifentanil	in vitro	↓ endothelial cell adhesion molecule; ↓ polymorphonuclear neutrophils migration;	[133]
Lei et al.	Remifentanil	in vitro	↑ cardiomyocytes protection against oxidative stress; ↓ PKCβ expression; ↓ autophagy activity;	[134]
Zhao et al.	Remifentanil	in vivo	↓ NF-kB expression; ↓ TNF-α expression; ↓ IAM-1 expression; ↑ protective effect against ischemia–reperfusion injury syndrome;	[135]
Hyejin et al.	Remifentanil	in vitro	↓ human neutrophils activations induced by lipopolysaccharide (LPS); ↑ anti-inflammatory effects;	[136]
Maeda et al.	Remifentanil	in vivo	↓ IL-6 expression in mouse brain; ↓ inflammation in cerebral tissue;	[137]
Hasegawa et al.	Remifentanil	in vivo	↓ inflammation expression caused by surgical stress;	[138]
Lu et al.	Ketamine	in vitro	↓ NMDA receptors activity; ↓ calmodulin-dependent protein kinase 2 (CAMK2) activity; ↓ NF-kB phosphorylation and nuclear translocation; ↑ protective against inflammation induced by LPS;	[139]
Wu et al.	Propofol	in vitro	↑ NMDA expression; ↑ Ca^2+^ accumulation; ↓ NF-kB phosphorylation; ↓ LPS- induced pro-inflammatory cytokine;	[140]
Inada et al.	Propofol	in vitro	↓ production of TNF-α and IL-6; ↓ production of neutrophil chemokines; ↑ anti-inflammatory effects;	[141]
Zhao et al.	Propofol	in vitro	↓ activity of NLR family pyrin domain containing 3 (NLRP3); ↓ inflammation and apoptosis mechanisms;	[142]
Plachinta et al.	Isoflurane	in vivo	↓ TNF-α activity; ↓ acidosis; ↓ damage to the vascular endothelium associated with LPS-induced inflammation; ↑ vascular protection during inflammation;	[143]
Wang et al.	Isoflurane	in vitro	↑ microRNA-9-3p expression; ↓ activity of fibronectin type III domain containing 3B (FNDC3B); ↑ protection against hepatic ischemia–reperfusion injury;	[144]
Lu et al.	Isoflurane	in vivo	↓ caspase-11, IL-1β and IL-18 expression; ↓ intracellular Ca^2+^ accumulation; ↑ protection against inflammation-induced by liver ischemia/ reperfusion injury;	[145]
Zhang et al.	Isoflurane	in vitro	↓ heme-oxygenase-1 (HO1) activity; ↓ redox, inflammation and apoptosis expression in sepsis- induced brain damage;	[146]
Lv et al.	Sevoflurane	in vivo	↑ neuro-inflammation and apoptosis by activation of cholinergic anti-inflammatory pathway;	[147]
Ngamsri et al.	Sevoflurane	in vivo	↓ neutrophil expression in peritoneal lavage; ↑ protection against peritonitis- induced sepsis in rats; ↑ expression of hypoxia- inducible factor 1α and adenosine A2B receptor in liver, intestine and lung;	[148]
Wang et al.	Sevoflurane	in vivo	↓ allergic airway inflammation; ↓ Th2 expression and NLRP3 activity;	[149]
Shen et al.	Sevoflurane	in vivo	↓ acute lung inflammation in ovalbumin-induced allergic mices; ↑ protection against airway remodeling in mouse; ↓ VEGF and TGF-β1 activity in lung tissues;	[150]
Wang et al.	Sevoflurane	in vivo	↑ inflammation of microglia in hippocampus of neonatal rats; ↓ activity of WNT/β-catenin/CaMKIV pathway;	[151]

**Table 2 cells-11-01880-t002:** Impact of perioperative antioxidant therapy on clinical outcomes in surgery.

Authors	Antioxidant	Study Description	Comments	Reference
Alshafey et al.	Vitamin C	- prospective randomized study; - included 100 patients; - vitamin C 2 g daily/3 days pre-operatively;	↓ incidence of atrial fibrillation; ↓ ventilator stay; ↓ need for inotropic support;	[243]
Gunes et al.	Vitamin C + α-tocopherol	- prospective study; - included 59 patients with cardio-pulmonary by-pass; - vitamin C 500 mg/day + α-tocopherol 300 mg/day, administered on the day of operation and for consecutive postoperative days;	↓ C-reactive protein (CRP) in patients who received antioxidant therapy; ↓ white blood cell count in control group; ↓ systemic inflammatory response;	[244]
Castillo et al.	Vitamin C + omega 3 polyunsaturated fatty acids (n-3PUFA)	- prospective study; - included 95 patients; - n-3PUFA 2 g/day administered 7 day before surgery; - vitamin C 1 g/day + vitamin E 400 IU/day administered 2 days before surgery;	↓ reduced/oxidized glutathione (GSH/GSSG) ratio; ↓ malondialdehyde (MDA) expression; ↓ lipid peroxidation; ↓ protein carbonylation; ↓ NF-kB activation;	[245]
Angdin et al.	Vitamin E + vitamin C + allopurinol + acetylcysteine	- prospective, randomized, double blind study; - included 22 patients;	↑ pulmonary vasodilatation protection; ↑ protection of endothelium-dependent vasodilation after cardio-pulmonary by-pass; ↓ endothelial dysfunction;	[246]
Antonic et al.	Vitamin C	- prospective randomized single center study; - included 100 patients; - vitamin C 2 g administered 24 h before surgery and 2 h before surgery; - vitamin C 1 g administered twice daily 5 days after surgery;	-no protective effects on the incidence of acute renal injury;	[247]
Das et al.	Vitamin C	- prospective study; - included 78 patients; - vitamin C 500 mg twice daily for 7 days prior to surgery;	↓ requirement of adrenaline; -no effects off length stay in ICU, and time of extubation;	[248]
Sadeghpour et al.	Vitamin C	- randomized clinical trial study; - vitamin C 2 g i.v. before surgery; - vitamin C 1 g daily oral for the first 4 postoperative days;	↓ hospital length of stay; ↓ intubation time; ↓ drainage volume in the ICU; ↓ inflammatory factors;	[249]
Samadikhah et al.	Vitamin C + atorvastatin	- randomized double blind clinical trial; - included 120 patients; - vitamin C 2 g/day in the surgery day + 1 g/day for 5 days postoperative + atorvastatin 40 mg	↓ atrial fibrillation incidence post coronary artery by-pass graft (CABG);	[250]
Dehghani et al.	Vitamin C	- prospective study; - included 100 patients; - vitamin C 2 g before surgery; - vitamin C 500 mg twice daily for 5 days after surgery;	↓ incidence postoperative atrial fibrillation; ↓ hospital stay; ↓ ICU stay;	[251]
Jouybar et al.	Vitamin C	- prospective study; - included 40 patients; - vitamin C 3 g administered 12–18 h before operation + vitamin C 3 g administered during surgery;	- no effects on IL-6 and IL-8 expression - no improvement on hemodynamics; - no effects on blood gas variables;	[252]
Papoulidis et al.	Vitamin C	- prospective study; - 150 patients enrolled; - vitamin C administered preoperatively and postoperatively;	↓ atrial fibrillation incidence; ↓ hospitalization time; ↓ ICU stay;	[253]
Collier et al.	Vitamin C + vitamin E + selenium	- prospective study; - antioxidant therapy administered 7 days from ICU admission for trauma patients;	↓ ICU stay; ↓ hospital stay;	[254]

## Data Availability

Not applicable.

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
