# Peer review of "The Impact of General Anesthesia on Redox Stability and Epigenetic Inflammation Pathways: Crosstalk on Perioperative Antioxidant Therapy"

_cells, 2022, doi:10.3390/cells11121880_

Round 1
Reviewer 1 Report
REVIEVER’S REPORT:
This review paper summarizes the research related to the impact of molecular and cellular mechanisms involved in inflammation induced by surgery and anesthesia in patients and the use of antioxidative therapies to minimize inflammation and complications post-surgery. The paper is generally well written and merits publication; however, the quality of the paper can be enhanced if the following points can be addressed.
1English Language and Style: There are many grammatical issues which need to be resolved before accepting this manuscript. For example:
There are several run-on sentences in the manuscript, which need to be revised. For example: Line 75-78
There are a few spelling and grammar mistakes which need to be checked, for example: Line 60 “immuno-inflammatory”, Line 106 - Exogenous
Other comments:
1 I suggest the authors to tabulate the analgesics used and their impact on inflammatory parameters. Similarly, it would be better if the authors can tabulate the list of antioxidants and their beneficial effects on surgery and recovery. This would make it easy for the readers to understand the advantages or limitations with each analgesic.
Reviewer 2 Report
Thank you for permitting me to review this manuscript
Introduction
at the end of the introduction , the primary objective of the review should be cited , the three last sentences are unnecessary and may be relocated in the discussion section
conclusions
lower opioid requirement after antioxydant therapy is dubious for me especially with the antioxydant displayed in this review
Redox disturbances should be defined before it appears in a sub chapter
figure 1 legnd
the pink rectangle need more explanation
such as ROS,?
Line 234 : transition from general anesthesia ventilation to RS is not clear
please provide reference
the OS in cardiac surgery , lung transplant and one lung ventilation is obviously not the same as a simple patient ventilation for other current surgeries not related to lungs as part of standard general anesthesia practice
Line 346 the immunosuppressive effects of morphine are mostly cited in laboratory findings not really in clinical practice as stated by authors themselves later in the same paragraph , therefore theses general statements without real clinical foundations should be avoided or at least
Line 350 PPR